# Modeling decision-making under uncertainty with qualitative outcomes

**Nachshon Korem**[1,2,3]*, **Or Duek**[1,3,4], **Ruonan Jia**[2], **Emily Wertheimer**[2,5], **Sierra Metviner**[2], **Michael Grubb**[2,6], **Ifat Levy**[2,5,7,8]

**1** Department of Psychiatry, Yale School of Medicine, New Haven, Connecticut, United States of America, **2** Department of Comparative Medicine, Yale School of Medicine, New Haven, Connecticut, United States of America, **3** U.S. Department of Veterans Affairs National Center for Posttraumatic Stress Disorder, VA Connecticut Healthcare System, West Haven, Connecticut, United States of America, **4** Department of Epidemiology, Biostatistics, and Community Health Sciences, Ben-Gurion University of the Negev, Be'er Sheva, Israel, **5** Interdepartmental Neuroscience Program, Yale University, New Haven, Connecticut, United States of America, **6** Department of Psychology, Trinity College, Hartford, Connecticut, United States of America, **7** Departments of Psychology and Neuroscience, Yale University, New Haven, Connecticut, United States of America, **8** WCu Tsai Institute, Yale University, New Haven, Connecticut, United States of America

* Nachshon.Korem@Yale.Edu

## Abstract

Modeling decision-making under uncertainty typically relies on quantitative outcomes. Many decisions, however, are qualitative in nature, posing problems for traditional models. Here, we aimed to model uncertainty attitudes in decisions with qualitative outcomes. Participants made choices between certain outcomes and the chance for more favorable outcomes in quantitative (monetary) and qualitative (medical) modalities. Using computational modeling, we estimated the values participants assigned to qualitative outcomes and compared uncertainty attitudes across domains. Our model provided a good fit for the data, including quantitative estimates for qualitative outcomes. The model outperformed a utility function in quantitative decisions. Additionally, we found an association between ambiguity attitudes across domains. Results were replicated in an independent sample. We demonstrate the ability to extract quantitative measures from qualitative outcomes, leading to better estimation of subjective values. This allows for the characterization of individual behavior traits under a wide range of conditions.

## Author summary

In the current study, we explored how people make decisions when the outcomes are not easily measured in numbers, such as with choices between medical treatments. Traditional mathematical models, which rely on numerical data, are not designed to handle such decisions, leading to a gap in understanding how people evaluate these qualitative outcomes. Using hierarchical Bayesian modeling, we developed a model that bridges this gap by translating qualitative outcomes into individualized quantitative values, enabling us to understand the underlying decision-making processes better. Our model not only provides a better fit to laboratory data than existing models with qualitative or

**Data availability statement:** Data and code are available on the author's GitHub. https://github.com/KoremNSN/QualMod https://github.com/KoremNSN/QualMod/tree/main/data.

**Funding:** This study was supported by the Yale Claude D. Pepper Older Americans Independence Center (c), and by NIH grants R21AG049293, R56AG058769, and NSF grant BCS1829439 to IL. The funders had no role in study design, data collection and analysis, decision to publish, or preparation of the manuscript.

**Competing interests:** The authors have declared that no competing interests exist

quantitative outcomes but also allows for meaningful comparisons of how people handle uncertainty across different decision-making scenarios. This approach opens new doors for studying decision-making in areas where traditional methods struggle, offering a more nuanced view of human behavior in complex situations.

## Introduction

Life is a series of decisions where most outcomes are uncertain. These decisions range from trying a new dish at our favorite restaurant to selecting a life-saving medical treatment. Often, these decisions are quantitative in nature; for example, buying a lottery ticket for $2 with a 1 in 300 million chance of winning $21 million. Many decisions, however, are qualitative, such as choosing between your regular coffee or paying more for premium coffee beans. The ability to compare different qualitative outcomes implies that we can derive some form of comparable subjective value for these outcomes [1]. Here, we aimed to quantify how qualitative outcomes affect uncertainty attitudes.

When presented quantitatively, uncertainty around choice outcomes can be categorized into two components: risk and ambiguity. Risk occurs when probabilities of potential outcomes are precisely known [2]; ambiguity refers to situations where these probabilities are partially or entirely unknown [3]. Prior research has shown that individuals generally exhibit an aversion to both risk and ambiguity in scenarios involving potential gains [4–7], but that these attitudes vary substantially across individuals and are not strongly correlated with each other [7–13]. Many studies have characterized these attitudes using choices with monetary (or point) outcomes [5,8,10,12,14–16]. While self-report questionnaires examined risk-taking across various domains [17], only a few studies quantified individual attitudes in non-monetary domains. Importantly, these studies still employed quantitative outcomes, including numbers of M&Ms and milliliters of water [1], months of extended lifespan [18], or milligrams of medication, and minutes spent with social partners [19]. Although some earlier studies did examine qualitative decision-making [20,21], there is still no straightforward way to model these kinds of decisions. Current models, such as cumulative prospect theory [22], are not practically equipped to handle qualitative outcomes. As a result, such decisions are typically not addressed in neuroimaging and psychiatric research.

In this study, we estimated risk and ambiguity attitudes in two separate modalities: quantitative (monetary decisions) and qualitative (medical decisions), leveraging computational modeling to extract values from qualitative outcomes and examine how uncertainty attitudes influence decision-making across various domains. Sixty-six in-person and 332 online participants engaged in a task of decision-making under uncertainty (see Table 1 for demographics), where they made choices between a certain outcome and the chance for a more favorable outcome. Our objectives were to (1) estimate subjective values for a range of qualitative outcomes, (2) assess the model's fit using quantitative outcomes for comparison, and (3) explore how attitudes towards uncertainty vary across different domains. Importantly, uncertainty in both domains was presented quantitatively (e.g., a 50% chance) rather than qualitatively (e.g., high chance). This allowed us to examine the subjective valuation of qualitative outcomes independently from qualitative risk representations.

## Results

To test our modeling approach to decision-making with qualitative outcomes, we analyzed choice data from 398 participants over 2 experiments (Table 1). Participants made a series of choices about risky and ambiguous options, which varied in their potential outcomes, the likelihood of obtaining the outcomes, and the level of ambiguity around these likelihoods

**Table 1. Demographic description of the sample.**

| | In-person sample N/ Mean (SD); min-max | Online sample N/ Mean (SD); min-max |
|---|---|---|
| Age | 49.14 (22.76); 18-88 | 49.49 (14.73); 20-80 |
| Sex (females) | 30 (45.45%) | 164 (49.40%) |
| MoCA | 28.39 (1.28); 26-30 | |
| Education | | |
| High school graduate, GED, or less | 4 | 27 |
| Some college or post-high school | 1 | 85 |
| College Graduate | 22 | 139 |
| Advanced graduate or professional degree | 27 | 44 |
| Estimated household income | | |
| $14,999 or less | 12 | 26 |
| $15,000-24,999 | 3 | 21 |
| $35,000-49,999 | 11 | 103 |
| $50,000-74,999 | 8 | 93 |
| $75,000-99,999 | 4 | 37 |
| $100,000-149,999 | 13 | 37 |
| 150,000-250,000 | 11 | 11 |
| $250,000-350,000 | 2 | 2 |
| $350,000 or more | 1 | |
| missing | 1 | 2 |
| History of major surgery | | |
| Yes | 29 | |
| No | 36 | |
| missing | 1 | |

(see Methods). The task was similar to a task used in multiple previous studies [12], with one critical difference: instead of the monetary outcomes of the original task, here, participants were presented with a hypothetical medical scenario (see Fig 1 and S1 Text) and had to make choices among different available treatments. The potential outcome of each treatment was described verbally (Methods) with no quantitative information (Note, however, that the uncertainty levels associated with each outcome were presented quantitatively). To allow for model validation in a quantitative domain, all participants also made choices about uncertain monetary options (Methods).

## Modeling approach

We employed hierarchical Bayesian modeling to capture and examine subjects' choice behavior. In the context of monetary decisions, we used a power utility function to model the subjective value assigned to each option [6] with a linear effect of ambiguity on the perceived probability (Equation 1 – Classic Utility Model) [12]. Hyperpriors were chosen based on previous data [12], with slight risk (mean 0.72) and ambiguity (mean 0.65) aversion (see S2 Text and S1 Table for comparison with less and uninformed hyper-priors). Importantly, the same priors were used for the two independent data sets.

Equation 1 – Classic Utility Model:

$$SV = \left(P - \beta \star \frac{A}{2}\right) \star V^{\alpha}$$

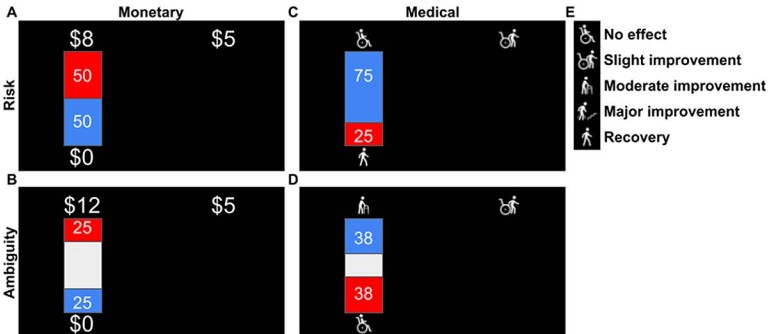

**Fig 1. Risk and ambiguity task.** Task Design: Participants were presented with choices between an uncertain option and a certain outcome across four scenarios: risky monetary decisions (A), ambiguous monetary decisions (B), Risky medical decisions (C), and ambiguous medical decisions (D). In the risky scenarios (A, C), the outcome probabilities were visually represented by red and blue rectangles, and these probabilities were fully disclosed to the participants. In the ambiguous scenarios (B, D), the probability information was partially obscured by a grey rectangle, indicating uncertainty. The outcome probabilities in risky trials were set at 25%, 50%, and 75%, while the levels of ambiguity (indicated by the grey area) were set at 74%, 50%, and 24%. There were four possible outcomes for monetary decisions ($5, $8, $12, and $25) and four potential medical outcomes (E). Each unique pairing of uncertainty and outcome levels was presented to the participants four times.

$$\alpha \sim Beta\left(\alpha_1, \alpha_2\right) * 2$$

$$\alpha_1 \sim Normal\left(4,1,0,\infty\right)$$

$$\alpha_2 \sim Normal\left(7,3,0,\infty\right)$$

$$\beta \sim Normal\left(\mu_\beta, \sigma_\beta, -1.5, 1.5\right)$$

$$\mu_\beta \sim Normal\left(0.65,1\right)$$

$$\sigma_\beta \sim Gamma\left(2, 1\right)$$

We modeled the subjective value (SV) using the objective probability (P; for risky options, P = 0.25, 0.5, or 0.75; for ambiguous options P = 0.5, and for the certain gain P = 1), ambiguity level (A; for risky or certain options A= 0; for ambiguous trials A = 0.24, 0.5, or 0.74), and potential winnings (V; 5, 8, 12, 25 dollars for the lotteries and 5 dollars for the sure bet). Risk attitude (α) and ambiguity attitude (β) were incorporated to capture individual differences in risk and ambiguity preferences.

To estimate the probability of choosing the lottery on each trial, we fitted a logistic choice function (equation 2), in which γ is the inverse temperature parameter.

Equation 2:

$$P_{risky} = \frac{1}{1 + exp^{\frac{\left(SV_{safe} - SV_{Risky}\right)}{\gamma}}}$$

$$\gamma \sim LogNormal\ (0, 0.25)$$

In addition, we tested "trembling-hand" logistic choice function (Equation 3), where there is less dependency between the risk and ambiguity parameters and the slope of the logistic function [23].

Equation 3 - Trembling-Hand Model:

$$P_{\text{Risky}} = \frac{1}{1 + exp^{-(SV_{safe} - SV_{Risky})}} * (1 - \delta) + \delta * .5$$

$$\delta \sim Beta(\delta_1,\ \delta_2)$$

$$\delta_1 \sim Normal(2, 1, 1, \infty)$$

$$\delta_2 \sim Normal(2, 1, 1, \infty)$$

In the medical domain, fitting the utility function is challenging due to the qualitative nature of the outcomes, which lacked a quantifiable value for the value (V) component of the model (although uncertainty was still presented quantitatively). To tackle this issue, we used the model to estimate the subjective value associated with each outcome. Notably, we excluded the risk aversion parameter, as subjective values were individually tailored to each outcome and participant (equation 4 – Estimated Value Model). Given the ordinal nature of the outcomes, the model employs a truncated normal distribution for each value. See S3 Text, S2 Table, and S1 Equation for a comparison with categorical models.

Equation 4 – Estimated Value Model:

$$SV = \left(P - \beta * \frac{A}{2}\right) * \sum_{n}^{n=1} \nu_i$$

$$\beta \sim Normal\ \left(\mu_\beta,\ \sigma_\beta, -1.5,\ 1.5\right)$$

$$\nu_i \sim Normal\ \left(\mu_{\nu_i}, \sigma_{\nu_i}, 0, \infty\right)$$

$$\mu_\beta \sim Normal\ (0.65, 1)$$

$$\sigma_\beta \sim Gamma\ (2,\ 1)$$

$$\mu_{\nu_i} \sim Normal\ (4,\ 2, 0,\ \infty)$$

$$\sigma_{\nu_i} \sim Gamma\ (3,\ 1)$$

In this model, we estimated the subjective value ($\nu$) of each outcome (i) based on the cumulative values of the preceding outcomes. We assumed the outcomes were ordinal (e.g., slight

improvement =<moderate improvement), and thus, we modeled the subjective value of each outcome as the sum of the subjective value of the preceding outcomes plus an additive value representing the improvement.

To evaluate the model's fit, we introduced a baseline model devoid of subject-specific parameters, serving as a 'straw man' to establish a reference point for the estimated model's performance. In the medical domain, we utilized the category level (e.g., 1 for slight improvement, 2 for moderate improvement) as the value (V) in the model (equation 5).

Equation 5 - No-Subjective Parameters:

$$SV = \left[ p - \left( \frac{A}{2} \right) \right] * \nu$$

## Extracting quantitative values from qualitative outcomes

We applied the Estimated Value model (equation 4) to the medical decision data to obtain estimated subjective values for the different levels of qualitative outcomes. For comparison, we also applied the No-Subjective Parameters model (equation 5) to the data (see Table 2 for model comparison). Using a Leave-One-Out (LOO) cross-validation method to estimate the

**Table 2. Model comparison.**

|  | Rank | LOO | p_loo | d_loo | Weight | SE |
|---|---|---|---|---|---|---|
| **In-person Sample** |  |  |  |  |  |  |
| **Monetary** |  |  |  |  |  |  |
| Estimated Value | 0 | -1563.26 | 225.65 | 0 | 0.96 | 45.63 |
| Classic Utility | 1 | -1910.72 | 142.76 | 347.56 | 0 | 42.37 |
| Trembling-Hand | 2 | -1976.14 | 138.06 | 413.26 | 0.04 | 36.68 |
| No-Subjective Parameters | 3 | -3457.45 | 33.94 | 1893.79 | 0 | 22.06 |
| **Medical** |  |  |  |  |  |  |
| Estimated Value | 0 | -1411.89 | 210.83 | 0 | 0.987 | 45.14 |
| No-Subjective Parameters | 1 | -2929.97 | 46.56 | 1518.02 | 0.013 | 32.17 |
| **Online Sample** |  |  |  |  |  |  |
| **Monetary** |  |  |  |  |  |  |
| Estimated Value | 0 | -3788.46 | 822.14 | 0 | 0.99 | 69.08 |
| Classic Utility | 1 | -5211.72 | 451.37 | 1423.26 | 0 | 57.64 |
| Trembling-Hand | 2 | -5798.14 | 404.43 | 2009.68 | 0 | 54.30 |
| No-Subjective Parameters | 3 | -11033.45 | 95.55 | 7244.99 | 0 | 17.68 |
| **Medical** |  |  |  |  |  |  |
| Estimated Value | 0 | -4102.47 | 787.08 | 0 | 0.99 | 72.65 |
| No-Subjective Parameters | 1 | -9514.81 | 181.52 | 5412.34 | 0.01 | 44.87 |

Rank: The position of a model based on its LOO (Leave-One-Out Cross-Validation) score, with 0 indicating the model with the best predictive performance. **LOO**: Expected log pointwise predictive density (ELPD), which measures how well a model can predict unseen data. A higher LOO score reflects better model fit and generalizability. LOO also accounts for model complexity to avoid overfitting—ensuring that more flexible models must demonstrate sufficiently improved predictive accuracy to justify their complexity. **p_loo**: Effective number of parameters - representing the model's complexity. Higher values indicate that the model has more flexibility in fitting the data. However, this is not a literal parameter count—it reflects the balance between the number of data points and how much the model adapts to capture meaningful patterns. The same model applied to a larger dataset would have a higher p_loo, as more data introduce additional variability to capture. **d_loo**: Difference in LOO between a given model and the best model (ranked 0). This shows how much worse the predictive performance is compared to the top-ranked model. **Weight**: Model weights based on the LOO scores, indicating the relative importance of each model. These weights help in model averaging, with higher weights given to models that show better predictive performance. **SE**: Standard error of the LOO estimate, reflecting the uncertainty in the score. Larger SE values indicate greater variability in the model's predictive accuracy across different data points.

out-of-sample predictive fit [24] (Methods), we compared the models. The Estimated Value model demonstrated a superior fit to the data in both samples. Subsequently, we extracted the mean added value for each category assigned by each participant. This approach provided estimates of the added values people attribute to the various outcomes. The mean value for slight improvement was 6.93, with a standard deviation (SD) of 1.79. This value increased by 9.00 (SD 2.62) for moderate improvement, by an additional 7.04 (SD 2.89) for major improvement, and finally by 4.18 (SD 2.41) for complete recovery (Fig 2A). For the online sample, the mean value for slight improvement was 8.63 (SD 2.54), which increased by 12.62 (SD 4.25) for moderate improvement, by an additional 4.66 (SD 2.56) for significant improvement, and finally by an additional 2.37 (SD 1.61) for complete recovery.

## Model validation and comparative analysis

We applied the Estimated Value model to the monetary decision data to facilitate a direct comparison with a classical utility model (equation 1). Using the monetary decision data, we evaluated the effectiveness of the Estimated Value model (equation 4) by comparing it to various models that incorporate an objective estimate of the outcome value. Specifically, using

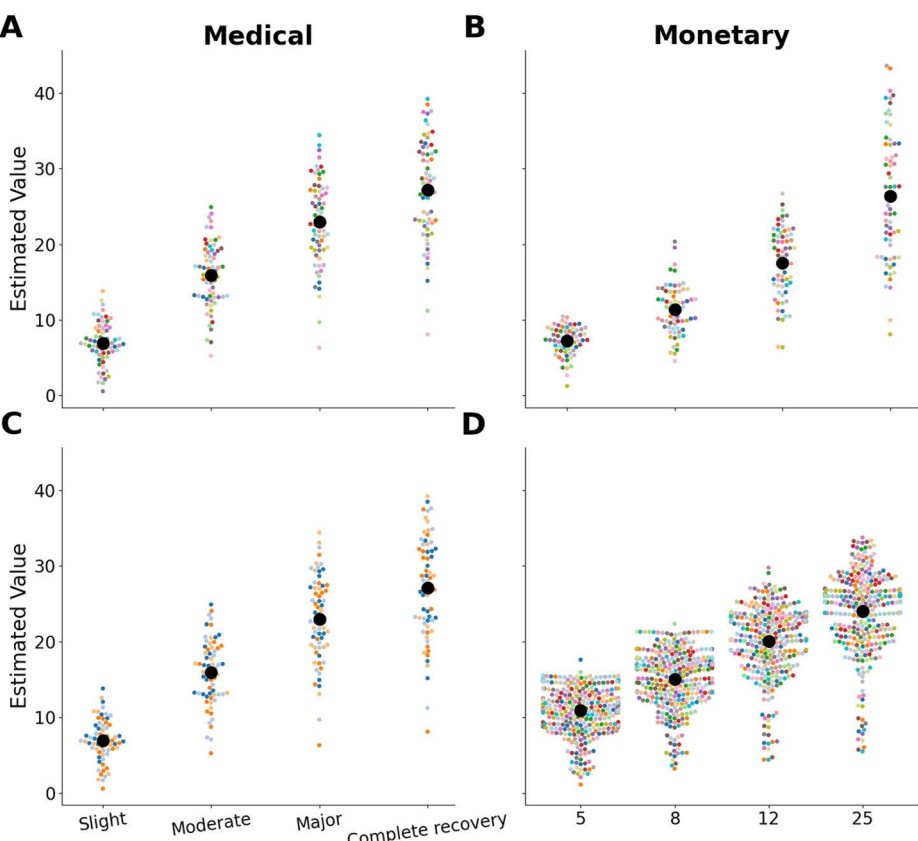

**Fig 2. Estimated value in each category.** The estimated values from the Estimated Value model for each category. Panes A and B show the estimated values for the medical decision-making task (Pane A) and the monetary decision-making task (Pane B) for the in-person sample. Panes C and D show the estimated values in the medical decision-making task (Pane C) and the monetary decision-making task (Pane D) for the online sample. Each colored dot represents an individual participant, while the large black dot indicates the mean estimated value for each category. See S4 Figure for an illustration of the curvature in the monetary domain.

LOO cross-validation, we compared the Estimated Value model against several alternatives: a No-Subjective Parameters model (equation 5), a Classical Utility model (equation 1), and a Classical Utility with a Trembling-Hand choice function model (equation 3). Our analysis revealed that the Estimated Value model outperformed the other models in terms of fit to the data within the monetary domain in both samples (see Table 2). This finding underscores the robustness of the Estimated Value model across different decision-making contexts.

We also derived values for the monetary categories. The mean value for $5 was 7.22 (SD 1.60). This value increased by 4.13 (SD 1.47) for $8, by an additional 6.23 (SD 2.18) for $12, and finally by 8.77 (SD 3.41) for $25 (Fig 2B). For the online sample, the mean value for $5 was 10.91 (SD 2.72), which increased by 4.13 (SD 1.84) for $8, by an additional 5.04 (SD 2.35) for $12, and finally by an additional 3.92 (SD 2.18) for $25.

## Simulations

The estimated model has more degrees of freedom. Hence, a better fit could result from over-fitting [25]. One way to ensure that the results represent real phenomena rather than overfit is by using simulations [26]. We simulated data (Methods) to assess the fit of the Estimated values model in scenarios where it should underperform (low noise) and overperform (high noise). The simulation results demonstrate how noise levels influence model performance. With lower noise (0.1), where the data generation function is closer to the utility curve, the classic utility model provided a better fit. However, as noise increased to 0.3 and 0.5, the estimated value model consistently showed superior performance, regardless of the number of participants. Specifically, at higher noise levels, the estimated value model had higher LOO values and weights across different sample sizes. For detailed results, see S5 Text and S3 Table.

## Cross-domain association between uncertainty attitudes

Finally, we explored how attitudes toward uncertainty vary across different domains. Using robust regression [27,28], we found that ambiguity aversion (β) in the medical domain was strongly and positively associated with the same attitude in the monetary domain (mean slope = 0.76, 89% HDP [0.63, 0.91]; Fig 3). This association was replicated in the online sample (mean slope = 0.37, 89% HDP [0.29, 0.44]). This finding suggests that attitudes towards

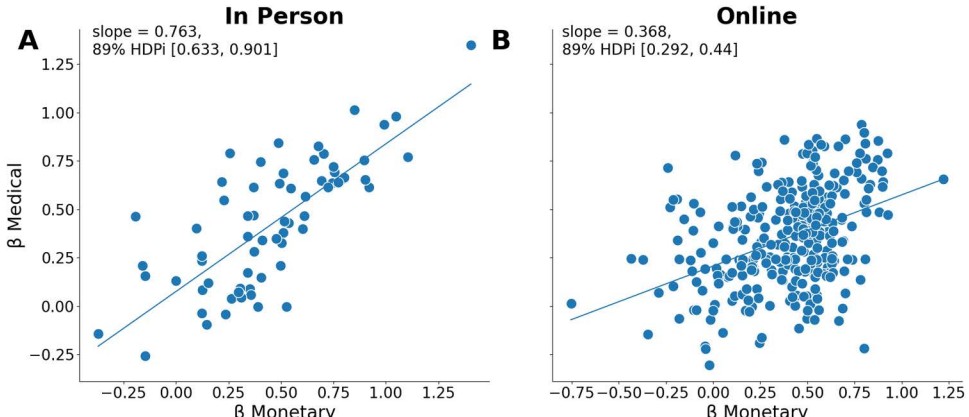

**Fig 3. Cross-domain association in ambiguity aversion (β).** The positive association between ambiguity aversion (β) in the monetary (x-axis) and medical (y-axis) domains in the in-person (Pane A) and online (Pane B) samples. The mean slope of the robust regression and the 89% highest density posterior interval (HDPi) are indicated in the panels.

ambiguity are consistent across different domains. Interestingly, the cross-domain association is stronger among individuals with a history of surgery (see S4 Text, S1, S2 and S3 Figs), suggesting that experience with potential outcomes may partially shape ambiguity attitudes. It is important to note that risk attitudes, as operationalized in our models, depend on the outcome levels; that is, the value incorporates the risk preference. Thus, it precludes a straightforward comparison of risk attitudes across domains.

## Discussion

This study aimed to quantify the values of qualitative outcomes and to characterize individual uncertainty attitudes when making choices between ordinal outcomes. Participants made a series of choices between certain outcomes and uncertain, potentially better, outcomes. Through computational modeling, we estimated the values participants assigned to different qualitative outcomes and assessed their attitudes towards ambiguity, which were consistent across two domains. Notably, our model outperformed a classical utility model with access to the objective amounts in the monetary domain. Overall, our model demonstrates a good fit and allows for better estimation of subjective values for both qualitative and quantitative outcomes.

When testing uncertainty attitudes, one challenge that often arises is how to treat qualitative outcomes [29]. Theoretically, this problem stems from the difficulty in comparing outcomes with unknown cardinal values [30]. Methodologically, it complicates the design of experiments and the interpretation of results, as traditional approaches often rely on quantitative measures [31]. Statistically, it poses challenges in modeling and analyzing data due to the lack of a consistent scale for comparison. To address this issue, researchers often quantify an aspect of the outcome (e.g., milligrams of medication) [18,19,32] or treat the outcomes as categories with a fixed distance between them [33]. In this study, we utilized computational modeling to extract estimated values for each outcome, providing insights into the processes underlying participants' decision-making. This approach allows for the use of different categories without the assumptions of a specific distance between them. By estimating these values, we can examine individual differences and use them in more precise parametric analyses, such as with neural representation of value, enhancing our understanding of the neural mechanisms involved in processing uncertainty. Nevertheless, while this model describes the observed data well, future applications should aim to extend its predictive capabilities to describe behavior more comprehensively. In particular, our experimental design relied on objectively ordinal outcomes (for example, moderate improvement is, by definition, better than slight improvement). A similar modeling approach could be used more generally for inferring values of options that are not inherently better or worse than each other (such as apples and oranges), but this should be empirically tested.

Uncertainty is especially relevant in the medical domain, as it is central to health decisions across the entire continuum of medical care [34,35]. While medical diagnoses and treatments are often described qualitatively, experiments assessing uncertainty attitudes typically quantify these outcomes into measures like years of life [18,36] or milligrams of medication [19]. This simplification abandons the original complex qualitative outcome. In contrast, our model allows for the introduction of complete qualitative outcomes. Nevertheless, our model still uses quantitative probabilities; in future research, it will also be interesting to include qualitative information about outcome likelihood (for example, "a high chance for success").

The field of healthcare, particularly medical decision-making, is a critical area of application, where ongoing debates address the level of information patients receive and how they

use it [35]. The fuzzy-trace theory posits that patients emphasize the gist representation of outcomes they are considering in decision-making despite also processing the verbatim representation simultaneously [37–39]. Our data can be viewed as an attempt to estimate this gist, with our Bayesian approach allowing us to preserve these values as distributions rather than forcing them into point estimates. This method captures a range of possible interpretations, providing a nuanced understanding of patient decision-making. Furthermore, the increased cross-domain association in ambiguity attitudes observed in individuals with prior surgical experience supports the transition from verbatim to gist processing [40]. This finding suggests that, similar to experts, personal experience may facilitate a transition toward greater reliance on gist-based processing, enriching our understanding of how experience shapes the processing of uncertainty in critical health decisions.

Beyond healthcare, our model can be applied in areas such as marketing, where product comparisons—particularly for new products—involve uncertainty. It offers a framework to quantify qualitative outcomes and enhance decision-making in contexts like product introduction [41,42]. For products with subjective attributes, such as comfort or quality, the model can capture consumer perceptions before purchase. For example, the value of the comfort of a new shoe can be modeled through elicited subjective preferences. Additionally, ambiguity aversion revealed in monetary decisions can influence preferences in other domains, such as a tendency to favor established brands over newer alternatives [43], particularly when attribute uncertainty is high. Our model can also adapt to scenarios with unknown probabilities [44]. In these cases, it will provide insight into decision rules by quantifying qualitative aspects and generating more informative payoff matrices.

Our focus on ambiguity - rather than risk - attitudes in this study warrants an explanation. Technically, in the Estimated Value model, risk attitudes are not quantified separately but are integrated into the estimated values (Equation 4). More broadly, the measure of risk attitude is tied to the outcome value. For example, a participant's preference for a 50% chance of $12 over a guaranteed $5 reflects how they value different monetary amounts. Similarly, choosing a guaranteed "slight improvement" over a 50% chance of a "moderate improvement" reveals how they value medical outcomes. In both cases, risk is inherently linked to the value of the outcome. In contrast, ambiguity attitudes are assessed by measuring how participants respond to uncertainty within the same domain, in relation to risk, independent of outcome value. This distinction allows us to compare the effect of ambiguity across different domains, such as monetary and medical decisions. Previous studies suggest that subtle experimental manipulations can alter decision attitudes [45] and that ambiguity attitudes are less stable over time [11,33] and lack a known structural correlate [46,47]. Nevertheless, our results indicate that ambiguity attitudes are consistent across domains when studied simultaneously. Additionally, we provide evidence that experience influences this association. Just as perceived risk shapes risk-taking behavior [17], perceived ambiguity can affect the subjective value of outcomes. Future studies should explore disentangling the risk component from the estimated values.

In the monetary domain, our Estimated Value model fits the data better than a classical utility model. While the utility model we used is constrained to follow a specific functional form [48], our estimated model is more flexible and can adapt to the data. Although models like cumulative prospect theory [22] and the Kőszegi-Rabin model [49] account for reference points, they assume stable relationships between outcome values along a continuous curve. These models do not capture how adding new options can shift the perceived value of existing choices, as seen in behavioral phenomena like the decoy effect [50,51]. In the decoy effect, introducing an additional option that is slightly inferior to an existing choice makes that choice more attractive. For example, if participants must choose between $5 for sure or

a chance for $25, adding a new $20 option could make the $25 option seem more appealing by comparison. These effects highlight the limitations of classical models, which assume that more data simply refine curve parameters without altering the relationships between outcomes. In contrast, our Estimated Value model can capture these categorical and contextual shifts, providing a more nuanced understanding of how people evaluate options in dynamic decision-making scenarios.

To mitigate the risk of overfitting, we employed a leave-one-out cross-validation procedure to compare the models. Our results emphasize that decisions traditionally considered quantitative, such as monetary choices, are better understood when accounting for subjective, qualitative aspects of the outcomes. The consistent performance of the Estimated Value model across both monetary and medical tasks suggests that individual value curves, even on ordinal scales, play a critical role in shaping decision-making. This finding supports the idea that qualitative elements underpin decision-making processes across domains, regardless of whether the outcomes are explicitly or implicitly defined.

Our simulations also support our modeling approach. We simulated hypothetical participants who made choices based on the Classical Utility Model. As expected, in scenarios with low noise levels, where the data generation function closely mimicked the utility curve, the Classical Utility model outperformed the Estimated Value model. This result highlights how the LOO algorithm effectively penalizes the Estimated Value model for its extra complexity. However, as noise levels increased, leading to a higher deviation from the utility function, the Estimated Value model consistently outperformed the Classical Utility model, regardless of the number of participants. This suggests that the Estimated Value model is more adaptable to noisy data, providing a better fit for real participants' behavior.

The estimated model may be better equipped to capture phenomena such as the framing effect or range-frequency theory [52–54]. Unlike the utility function, which assumes outcomes are sampled from a single curve describing a person's behavior, the estimated model allows for unique curvatures for each set of outcomes. This means that the lowest and highest amounts create a frame of reference against which all other outcomes are compared. By learning about the possibilities within this frame, participants adjust their expectations accordingly. This flexibility enables the estimated model to more accurately reflect how people perceive and evaluate different outcomes in varying contexts.

Having both monetary (quantitative) and medical (qualitative) datasets allowed us to assess the Estimated Value model with quantitative data and gain confidence in the model's ability to extract values in the qualitative dataset. The model's fit on quantitative data provides evidence that the estimates for the qualitative outcomes represent participants' true values. While these estimates are on a relative scale, without specific measurement units, they open the possibility for use in future studies to examine value representation in the brain. Additionally, these estimates can be applied to evaluate the values of outcomes across different scenarios. Using the categorical variant, the model can compare discrete outcomes (e.g., comparing apples to oranges, choosing between an apple, or a 50% chance of two oranges). Overall, enhancing our understanding of how people perceive and compare various types of outcomes.

To conclude, we present a model capable of assigning quantitative values to qualitative outcomes. The model demonstrates a better fit for both qualitative and quantitative data compared to other potential models. Although more complex than a classical utility function, both model comparisons and simulations suggest that the improved fit is not due to overfitting. This model opens new avenues for exploring the relationships between different domains, outcomes that cannot be objectively quantified, and the representation of value in the brain.

## Methods

### Study 1 (in-person)

**Ethics statement.** The study was approved by the Yale Human Investigation Committee 0910005795 and followed institution guidelines.

**Participants.** A sample of sixty-six individuals with valid data was drawn from one hundred and one adults (48 females; age range = 18–89; mean 52.97, SD ±22.41) who were screened for the experiment. All participants were screened over the phone to ensure the absence of major medical conditions, including neurological illness and lifetime Axis I psychiatric disorders. Participants provided written consent after a detailed explanation of the study. Ten participants did not complete the study and were not included in the analysis, resulting in a sample of ninety-one participants (failed to complete the task n=4; failed to come to consecutive sessions n=6).

To ensure all participants were cognitively healthy, we administered the Montreal Cognitive Assessment (MoCA) [55], which can detect mild cognitive impairments. Data from participants who scored less than 26 on the test were excluded [56], resulting in a sample of seventy-one cognitively healthy participants (32 females; age range = 18–88; mean 49.68 ±22.3 SD).

**Procedure.** Participants in this study came for three sessions completed within one week. All task data reported here were collected on the first session. The MoCA was completed in session 3, along with several other questionnaires. In brief, in session 1, participants completed the decision-making task (reported here) and a reversal reinforcement learning task (not reported here). In session 2, participants completed an fMRI task. Finally, in session 3, participants completed several questionnaires assessing cognitive ability and general IQ. Participants were paid for each session separately and received extra compensation for successfully completing the experiment.

**Risk and ambiguity in the monetary domain.** The task was based on a previously developed task [12] used in multiple studies. On each trial, participants chose between a small certain gain ($5) and a lottery that offered a larger amount. The lottery was risky in half of the trials, i.e., with known outcome probability. The risky lotteries were represented as bags containing red and blue chips. The numbers of red and blue chips were indicated by the percentage of a rectangle colored in red and blue and the numbers on the bag. Three different outcome probabilities were used (25%, 50%, and 75%). Dollar amounts ($5, $8, $12, and $25) next to each color (Fig 1a) indicated the amount of money that could be won if a chip of that color was drawn. In the remaining trials, the lottery was ambiguous, i.e., outcome probability was not precisely known. Ambiguity was achieved by occluding part of the bag (Fig 1b), rendering the probability of drawing a chip of a certain color partially unknown. Increasing the occluder size (24%, 50%, and 74%) increases the ambiguity level or the range of possible probabilities for drawing a red or blue chip. Each combination (amount, risk/ambiguity) was repeated four times. On twelve (out of 84) trials, participants were asked to choose between $5 for sure and a chance to win $5. Those trials were used as attention checks. Participants who failed six or more attention checks (n=3) were removed from the analysis in the risk and ambiguity task. In addition, participants who chose the lottery less than two times were omitted from the analysis because their data could not be fitted with a model (n=2). This exclusion was necessary because our choice function requires response variability; a consistent choice of one option provides no data points for model estimation. The final sample included 66 participants. At the end of the experiment, the computer randomly selected one of the trials, and the participants acted out the trials by selecting a chip. However, they did not receive the additional payment. We opted for a hypothetical outcome to make the monetary and medical conditions similar to each other (see below).

**Risk and ambiguity in the medical domain.** Participants were presented with a hypothetical scenario in which they were involved in a car accident and, as a result, suffered a spinal injury (S1 Text for more details). They were asked to choose between two medical treatments (Fig 1c and 1d), a known treatment with a known outcome ("slight improvement," parallel to a fixed monetary gain of $5) or the experimental treatment (Fig 1e), with outcomes varying in the level of improvement ("moderate," "major," or "complete recovery"). To align with the monetary task, these outcomes were created on an ordinal scale. The likelihood of the outcome of the experimental treatment varied with different levels of risk and ambiguity (parallel to playing a lottery). Outcome probabilities and ambiguity levels were the same as in the monetary task and were presented graphically and verbally. All aspects of the experimental design were similar to the monetary task. The order of the monetary and medical tasks was counterbalanced across participants.

## Study 2 (online)

This dataset was used to confirm and replicate the results. This is a secondary analysis of a previously published dataset [33].

**Participants.** Valid data from three hundred and thirty-two out of four hundred and four adults (212 females; age range = 20–80; mean 49.362, SD ±14.849) who were recruited using Amazon Mechanical Turk (mTurk) was analyzed. Participants provided consent online after reading a detailed explanation of the study following Yale Human Investigation Committee guidelines. Seventy-two participants did not complete the study and were not included in the analysis, resulting in a sample of three-hundred and thirty-two participants.

**Procedure.** Participants completed tasks similar to those described above, following the same procedures as the in-person sample, with two exceptions [33]. First, each lottery was presented twice instead of four times. Second, an additional condition of 100% ambiguity was introduced.

**Data simulations.** By controlling the data generation function, we assessed the performance of different models. If the estimated model outperforms the utility function, even when the data were generated by the utility model, it suggests that the estimated model is insufficiently penalized for its complexity, indicating a risk of overfitting. Using the utility function (Equation 1), we simulated datasets with varying parameters, controlling for noise levels and sample sizes. Each selection in the simulation included noise drawn from a normal distribution with a mean of 0 and a standard deviation determined by the simulation. Additionally, each subject had unique **risk (α) and ambiguity (β)** parameters. To ensure model convergence, we constrained risk attitudes to the range [0.1, 1.6] and ambiguity attitudes to [-1.4, 1.4]. We tested the following sets of parameters (N, noise): (30, 0.1), (30, 0.3), (30, 0.5), (60, 0.3), (60, 0.5), (120, 0.5), (300, 0.5). We then evaluated the utility function and the estimated value model (Equation 4) using **LOO scores and weights** to compare their ability to fit the simulated data.

## Hierarchical Bayesian modeling

By leveraging hierarchical Bayesian modeling (HBM), we were able to uncover hidden variables that offer valuable insights into the underlying mechanisms of decision-making. HBMs allow for partial pooling of data across the population, meaning that individual data points contribute to both individual and group-level estimates. This results in more robust and accurate posterior distributions compared to non-hierarchical models, particularly when dealing with small sample sizes or individual-level variability [57]. A key element in HBM is hyper-priors. Hyperpriors are higher-level distributions that set prior beliefs on the parameters of the model's primary prior distributions. They are beneficial as they allow for incorporating prior knowledge and uncertainty about the parameters, leading to more flexible and

robust models. Overall, this method allowed us to compare different models and assess which model described the data better.

### Leave-one-out cross-validation

This method splits the data into training and testing datasets. The model is trained on the training data and evaluated on the held-out test. This process is done repeatedly. We used the 'Arviz' implementation to compute the LOO of the models. Unlike Log Likelihood, LOO measures expected log pointwise predictive density (ELPD). Thus, higher LOO points to a better fit. Additionally, LOO inherently accounts for model complexity through the effective number of parameters, ensuring that models are not rewarded for overfitting.

### Model convergence

All models converge with rHat<1.01 and effective sampling rate > 1000. All analyses were conducted in Python 3.10.14, utilizing the 'PyMC' (version 4.1.7) [58] and 'ArviZ' (version 0.17.1) [59] packages. We utilized the No-U-Turn Sampler (NUTS) for Markov chain Monte Carlo (MCMC) inference, adhering to PyMC's default settings: 1000 draws, 1000 tuning steps, no thinning, and an 80% acceptance rate. The code can be found here: https://github.com/KoremNSN/QualMod/tree/main

## Supporting information

**S1 Text. Hypothetical medical scenario.**
(DOCX)

**S2 Text. Sensitivity analysis priors.**
(DOCX)

**S1 Table. Model comparison sensitivity analysis priors.**
(DOCX)

**S3 Text. Sensitivity analysis categorical.**
(DOCX)

**S1 Equation. Full categorical model.**
(DOCX)

**S2 Table. Model comparison sensitivity analysis categorical.**
(DOCX)

**S4 Text. Surgery analysis.**
(DOCX)

**S1 Fig. Posterior distribution of ambiguity attitudes for participants with and without a history of surgery.**
(TIF)

**S2 Fig. Cross-domain associations of ambiguity attitudes (β) in monetary and medical domains.**
(TIF)

**S3 Fig. Posterior distribution of cross-domain slopes for participants with and without a history of surgery.**
(TIF)

**S4 Fig. Estimated values in relation to actual monetary sums.**
(TIF)

**S5 Text. Simulation analysis.**
(DOCX)

**S3 Table. Model comparisons simulation analysis.**
(DOCX)

## Author contributions

**Conceptualization:** Or Duek, Ruonan Jia, Sierra Metviner, Ifat Levy.

**Data curation:** Nachshon Korem, Emily Wertheimer, Sierra Metviner.

**Formal analysis:** Nachshon Korem, Or Duek, Michael Grubb.

**Funding acquisition:** Ifat Levy.

**Investigation:** Nachshon Korem, Or Duek, Michael Grubb, Ifat Levy.

**Methodology:** Nachshon Korem, Or Duek.

**Project administration:** Nachshon Korem, Ifat Levy.

**Resources:** Nachshon Korem, Or Duek, Ruonan Jia, Michael Grubb.

**Software:** Nachshon Korem, Or Duek.

**Supervision:** Nachshon Korem, Ifat Levy.

**Validation:** Nachshon Korem, Or Duek, Michael Grubb.

**Visualization:** Nachshon Korem.

**Writing – original draft:** Nachshon Korem.

**Writing – review & editing:** Nachshon Korem, Or Duek, Ruonan Jia, Sierra Metviner, Michael Grubb, Ifat Levy.

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
