## [Decision Letter · Decision Letter 0]

9 Oct 2024

Dear Dr. Korem,

Thank you very much for submitting your manuscript "Modeling Decision-Making Under Uncertainty with Qualitative Outcomes" for consideration at PLOS Computational Biology.

As with all papers reviewed by the journal, your manuscript was reviewed by members of the editorial board and by several independent reviewers. In light of the reviews (below this email), we would like to invite the resubmission of a significantly-revised version that takes into account the reviewers' comments.

We cannot make any decision about publication until we have seen the revised manuscript and your response to the reviewers' comments. Your revised manuscript is also likely to be sent to reviewers for further evaluation.

Sincerely,

Varun Dutt, Ph.D

Academic Editor

PLOS Computational Biology

Zhaolei Zhang

Section Editor

PLOS Computational Biology

Reviewer's Responses to Questions

**Comments to the Authors:**

Reviewer #1: Intro (and discussion): It would be helpful to be more precise in the description of what is considered qualitative. First, the uncertainty is still expressed numerically and therefore quantitatively. Second, while the description of the medical outcomes are qualitative, the outcome variable still has an exogenously determined quantitative “objective” ordering, so sidesteps the challenge of finding a common metric mentioned on p25. Specifically, the stimuli use a quantitative scale, imposing a translation to a quantitative stepped measure for the participant (which allows for the model in Eq 5, and facilitates extracting subjective values). This can be contrasted with studies on food preferences which may not have a prior orderings. Those studies are often designed to first measure individually defined subjective preferences for foods and then convert those ratings into a quantitative estimate/common currency with more flexibility. It also allows for indifference. This work could also be contrasted with choices between two medications that led to different types of side effects, and the question of whether it’s similarly possible to estimate subjective values in that case.

Literature : To emphasize the usefulness of making cross-domain comparisons of ambiguity, it seems helpful to cite evidence for the benefits of such approaches as shown in cross-domain variance in risk (e.g., Blais, A. R., & Weber, E. U. (2006). A Domain-Specific Risk-Taking (DOSPERT) scale for adult populations. Judgment and Decision making, 1(1), 33-47.) It may also be helpful to mention research comparing novel (e.g. uncertain) to known (e.g. certain) products and/or brands as examples of research on qualitative ambiguity, since they could be considered analogous to the certainty-equivalent paradigm in the present work.

Expt 1 Methods: There are a range of exclusions described sequentially throughout the methods section. Please have a clear statement of the initial sample size and final sample size (e.g., 66) in the section on participants. Similarly, while transparency on the full procedures is appreciated, it would be useful to highlight which data (taken from which of the three sessions) was specifically used for this research. For example, am I correct that the data used here was not mixed across behavioral and scanner sessions?

Expt 2 Methods: Please provide a brief description of points of distinction between E1 and E2. (It feels odd to have to hunt up a preprint to understand the similarity of the methods.)

Modeling results : It seems notable that the Estimated Value model was ranked above the No-subjective Parameters in both the monetary and medical tasks similarly. In some sense, that highlights the importance of modeling the individual difference value curves across an ordinal scale regardless of whether that scale is implicitly or explicitly defined, correct? It seems the paper’s framing could be flipped, in that it implies that even seemingly quantitative decisions are better understood through modeling their qualitative aspects.

Results / Figure 2: The way that the medical and monetary findings are plotted may be misleading. The medical outcomes do have a step scale that’s faithfully reported as fixed intervals on the x-axis in panels A/C. However, it seems odd to have monetary increments of differing amounts plotted in similarly spaced intervals in panel B/D, and this also hides a nice feature of the data, which is an apparent diminishing marginal return as the amount of money increases. A more worrisome interpretation though, is that the linear trend across all four panels could show that participants simply coded both types of stimuli as four levels/steps of outcomes.

Results : Given that participants in Experiment 1 could be roughly split by whether or not they had a history of major surgery, it would be very useful to compare the correlation between the medical and monetary ambiguity coefficients between these groups to better speak to the potential qualitative comparison across domains.

Results : Healthwise, there seems to be a natural psychological/categorical break between “needs a wheelchair” and “doesn’t need a wheelchair” – is that visible in the data? In addition, to better understand the behavioral baseline, what was the estimated valuation for “no effect” as per the model estimates or the data itself?

Discussion/contribution : Summarizing some of the points made above – the benefit of the subjective modeling is clear, and useful. Overall, the model and data here offer important findings, but may be more limited in what they show than what is claimed in the introduction and discussion. This is primarily because the medical stimuli are not wholly qualitative, but rather coded, described, and presented in monotonic increasing steps. As such, it would be helpful to have a discussion of how much they do or do not advance our understanding of qualitative value over the monetary amounts, which clearly have their own qualitative subjective value. While this may be one step towards more qualitative models, it may not be able to account for less linear or well defined domains, or outcomes with mixed-valence attributes contributing to their values.

Reviewer #2: The authors apply sophisticated computational modeling to decision making under risk and ambiguity, comparing alternative models. There are a number of notable results, among them that qualitative outcomes can be modeled using techniques also applied to quantitative outcomes and that ambiguity attitudes generalize across very different domains (money and medical outcomes). The choice of medical outcomes is especially important (degrees of recovery due to standard treatment, slight improvement, compared to moderate/major/complete recovery). Similar results are obtained across independent samples (online and in-person). The methods are rigorous (e.g., counterbalancing quantitative and qualitative decisions, etc.).

The main issue with the current version of the manuscript is that the mechanisms underlying the models are not emphasized sufficiently. The authors do not sufficiently connect the models here to the decision-making literature in terms of the claims and findings in that literature. For example, what are the theoretical assumptions of this version of the classic utility model, how do these models relate to prospect theory, etc. Also, walking the reader through some well-chosen concrete examples would greatly help readers appreciate what it means to support alternative models. The authors should also give the reader a sense of the major progress of ideas in decision theories based on empirical tests. The author should explain how the current modeling results cohere with the larger decision-theory literature.

In addition, the modeling could be made much more accessible to a broader range of readers. Although this is a computational journal, the potential readership for this work is far broader than those who are highly familiar with this approach to modeling. For example, what do the 225.65 parameters of the estimated value model represent—how are they interpreted (Table 2)?

Similarly, regarding simulations, it is not clear how the reader should take the message that one model is best (classic utility) but another model (estimated value) showed superior performance as noise increased.

Overall, good models should explain as well as “predict” (simulate) out of sample; the best models also make predictions to new situations (ideally counterintuitive predictions that distinguish explanations) that reveal underlying mechanisms. How do the current models map onto the aforementioned desirable features? (What are the “insights into the processes underlying participants’ decision-making?”) What are novel predictions from the supported models? What do the supported models tell us that is new about the brain or the psychology of decision-making? In this connection, being able to model qualitative outcomes is quite an accomplishment, and it seems to map onto recent theory that distinguishes qualitative gist representations from precise quantitative representations—how do the current results speak to those relevant constructs?

The first two paragraphs of the Discussion are very good. The rest is good too but needs to be developed a bit more: “Without the assumptions of specific relationships among them” undersells this strength of the research; the methods here allow the relationships to be discovered rather than assumed.

The authors need to explain more about why risk attitude measures “require equating the outcomes across domains” but ambiguity measures do not, given how similarly they are operationalized.

What does “follow a specific curve” mean? There are many ways that could be interpreted.

Naturally, having more degrees of freedom is a double-edged sword; why is that better here in light of prior findings?

Real participants’ data are not necessarily more noisy in the sense of random fluctuation so it is not clear why simply being robust to more “noise” (as opposed to systematic variation) is good.

The authors make a good point that “This result highlights how the LOO algorithm effectively penalizes the Estimated Value model for its extra complexity.” On the one hand, this raises the issue that models should not be penalized for parameters that other work shows must be included to account for specific results. On the other hand, accommodating these parameters could distort results for needed parameters in a specific dataset. The authors should bring such considerations to bear (about penalizing models for assumptions shown to matter in other research) throughout in evaluating the models.

The authors make a couple of excellent points about the estimated model allowing for unique curvatures for each set of outcomes. The authors need to give examples of empirical effects that require that assumption and relate this statement to specific prior models (e.g., cumulative prospect theory, Kőszegi-Rabin, etc.). Concrete examples of decision problems and findings help.

Excellent measurement points are made in the last two paragraphs.

In summary, this is a sophisticated modeling approach to prediction that accommodates different kinds of uncertainty, which provides ways to quantify qualitative outcomes and to compare ambiguity attitudes across monetary and medical outcomes. These are impressive accomplishments, but they will not be accessible to, and sufficiently appreciated by, many readers if not explained in light of the prior literature and mechanisms as discussed above.

**Have the authors made all data and (if applicable) computational code underlying the findings in their manuscript fully available?**

Reviewer #1: None

Reviewer #2: None

PLOS authors have the option to publish the peer review history of their article (what does this mean? ). If published, this will include your full peer review and any attached files.

**Do you want your identity to be public for this peer review?** For information about this choice, including consent withdrawal, please see our Privacy Policy .

Reviewer #1: No

Reviewer #2: No
---

## [Decision Letter · Decision Letter 1]

25 Dec 2024

PCOMPBIOL-D-24-01428R1

Modeling Decision-Making Under Uncertainty with Qualitative Outcomes

PLOS Computational Biology

Dear Dr. Korem,

Thank you for submitting your manuscript to PLOS Computational Biology. After careful consideration, we feel that it has merit but does not fully meet PLOS Computational Biology's publication criteria as it currently stands. Therefore, we invite you to submit a revised version of the manuscript that addresses the points raised during the review process.

Please submit your revised manuscript within 30 days Feb 24 2025 11:59PM. If you will need more time than this to complete your revisions, please reply to this message or contact the journal office at ploscompbiol@plos.org. Please include the following items when submitting your revised manuscript:

We look forward to receiving your revised manuscript.

Kind regards,

Varun Dutt, Ph.D

Academic Editor

PLOS Computational Biology

Zhaolei Zhang

Section Editor

PLOS Computational Biology

**Journal Requirements:**

1) We have noticed that you have uploaded Supporting Information files, but you have not included a list of legends. Please add a full list of legends for your Supporting Information files after the references list.

2) We note that your Data Availability Statement is currently as follows: "Data will become available upon publication. All analysis code is available on the author's GitHub.". Please confirm at this time whether or not your submission contains all raw data required to replicate the results of your study. Authors must share the “minimal data set” for their submission. PLOS defines the minimal data set to consist of the data required to replicate all study findings reported in the article, as well as related metadata and methods (https://journals.plos.org/plosone/s/data-availability#loc-minimal-data-set-definition).

- The points extracted from images for analysis..

3) Please ensure that the funders and grant numbers match between the Financial Disclosure field and the Funding Information tab in your submission form. Note that the funders must be provided in the same order in both places as well.

**Reviewers' comments:**

Reviewer's Responses to Questions

**Comments to the Authors:**

Reviewer #1: 1. To re-state a point from the first review – the design of the experiment requires a step that translates the qualitative experience on to a quantitative monotonic linear scale. This appears useful here and the findings are compelling. But the methods and contributions of the work should then be represented more precisely. The model in the paper still uses (relies on?) “numerical data” just like the description of traditional models in the authors summary and the risk papers cited in the introduction. (In the discussion, sentence 208 qualifies that the paper is about choices between “ordinal outcomes.” These came from the simplified recovery scale, but the subsequent paragraph argues against oversimplified quantifications of qualitative outcomes, which is an odd juxtaposition.) My overall conceptual concern is that the ability to compare qualitative fields is still sometimes described in the paper as a contribution arising specifically from the *modeling* ( e.g., “leveraging computational modeling to extract values from qualitative outcomes”) rather than also the gathering of quantified data in the experiment and/or overall research. This needs to be described more carefully to understand future applications and generalization.

2. Perhaps a facet of the above issue, although there the paper uses ordinal outcome data, the illustrative example of what the model could do in the introduction is nominal (Thai vs. Italian). It’s less clear to me that this broader claim is sufficiently demonstrated in the present work. Indeed, the response letter describes nominal (apples/oranges) data as a future application. The absence of a need for fixed distances in the measurements certainly speaks to the feasibility of categorical/nominal applications, as do the categorical models in the supplement, but there is not *evidence* demonstrated in the paper for it. If the broader claim about all qualitative data is made, it seems reasonable to expect findings that speak to it. To be clear, I find the paper compelling. I also feel it is critical to be precise in terms of what this specific work has proven/shown versus what it shows capability/ feasibility/ promise to do.

3. From the values reported in the results section, it seems that the recovery data also shows diminishing marginal value as the scale increases (assuming linearity of the underlying scale), similar to the monetary data. Is that correct? Note the question in the prior review about how participants might *interpret* the scale remains regardless of whether an expert designed it to be linear. Would it be useful to comment on that parallel subjective structure between the monetary and health perceptions?

4. I’m a little confused by the statement that cumulative prospect theory is not equipped to handle qualitative outcomes. Is that meant literally in terms of the equations (rather than conceptually/theoretically)? For example, prospect theory (and thus presumably CPT) predicts diminishing marginal returns in subjective value, which seems consistent with what was observed in the qualitative recovery data (see prior question)? I apologize if I’m missing some context to this argument- it seems like integrating these findings with prior theory would be a positive benefit.

5. Upon re-review, and with the addition of discussion of products, the authors should probably cite the following paper : Muthukrishnan, A. V., Luc Wathieu, and Alison Jing Xu. "Ambiguity aversion and the preference for established brands." Management Science 55, no. 12 (2009): 1933-1941. As the title implies, the paper shows that ambiguity aversion in financial gambles correlates with ambiguity aversion in brands (e.g. preferring established brands vs. newer ones.)

6. The finding that experience with surgery improves cross-domain association seems to be an important theoretical point. (This was more of a comment than a question. But despite not being central to the computational model, might it be valuable to move that analysis to the main paper?)

Reviewer #2: The authors have produced a responsive revision, one that retains the considerable strengths of the original version, but the message is more accessible and more impressive. The strengths include applying sophisticated computational modeling to decision making under risk and ambiguity, comparing alternative models. There are noteworthy results, among them that qualitative outcomes can be modeled using techniques also applied to quantitative outcomes and that ambiguity attitudes generalize across very different domains (money and medical outcomes). The choice of medical outcomes is especially important (degrees of recovery due to treatment). Similar results are obtained across independent samples (online and in-person). The methods are rigorous (e.g., counterbalancing quantitative and qualitative decisions, etc.).

The authors have addressed the issues of explanation as opposed to description. Their responses have enhanced and clarified their message. The Discussion especially is greatly enhanced, and it now flows beautifully and communicates important conclusions. The revised manuscript is well above threshold—except for a couple of accuracy issues that are easily addressed; see below. However, although the modeling fits within the tradition of psychophysical approaches, I hope that the authors continue to think about explanation. The authors make an excellent case that their models describe a wide range of observed phenomena, more than satisfying the technical requirements of modeling. With powerful and flexible models, it is possible to describe a range of behaviors—and this is no mean feat—in terms of these psychophysical parameters, but the extent to which these parameters actually explain (as contrasted with describe) these behaviors remains a question not only for this work but for much contemporary work on decision making. Predicting decoy and other effects is fantastic, but it is predicting known results; ideally, the best theories predict new, counterintuitive effects. In addition, beyond these data, the authors do not account for some findings that dispute these psychophysical models, but that is beyond the scope of the present work. The authors should continue to think about these issues, but they are operating in a venerable tradition nevertheless.

As for the accuracy issues, first, the use of the word “only” is a major problem in the following sentence: “The fuzzy-trace theory posits that patients often hold only a gist representation of the outcomes they are considering [37].” To be clear, fuzzy-trace theory does not say that only gist is considered. On the contrary, fuzzy-trace theory says that multiple gist representations and a verbatim representation are encoded (and stored and retrieved) in parallel (e.g., see Reyna, 2012; Reyna, Müller, & Edelson, 2023). However, the tendency to lean more on the simplest gist representation accounts for many empirical phenomena. Because the gist representation is ultimately determinative of behavioral phenomena (e.g., the response pattern in gain-loss framing, the Allais effect, etc.) does not mean that it is the only representation that is processed. Metaphorically, gist is often (for theoretically specified reasons; see papers above) the deciding vote but it is not the only vote, and all of the processes in fuzzy-trace theory contribute to behavior, not just the simplest gist. The following would be accurate and can be substituted for the problem sentence with “only”: “The fuzzy-trace theory posits that patients emphasize the gist representation of outcomes they are considering in decision-making, despite also processing the verbatim representation simultaneously [37].”

The second accuracy correction per p. 30: “Furthermore, the increased cross-domain association in ambiguity attitudes observed in individuals with prior surgical experience supports the transition from gist to verbatim processing [38].” The transition in the cited paper is exactly the opposite, from relying more on verbatim processing (in novices) to relying more on gist processing (in experts). Developmental data as well as data on novice-expert transitions supports the direction from verbatim to gist. The authors’ results discussed here also support this direction from verbatim to gist. So simply reversing the phrases works to correct this problem, as follows, which is probably what the authors meant: “Furthermore, the increased cross-domain association in ambiguity attitudes observed in individuals with prior surgical experience supports the transition from verbatim to gist processing [38].”

However, this then does present a problem for the sentence that follows: “This finding suggests that personal experience may reinforce a shift towards more detailed, quantitative representations, enriching our understanding of how experience shapes the processing of uncertainty in critical health decisions.” The authors should consider whether their results support this sentence (I think not) and it does not agree with other developmental research on experience/expertise. They should edit the sentence in some way, otherwise it is a non sequitur, or simply delete it.

Reviewer 1’s points were insightful and the authors’ responses to them enhanced the manuscript. Per the reviewer’s comment “To emphasize the usefulness of making cross-domain comparisons of ambiguity, it seems helpful to cite evidence for the benefits of such approaches as shown in cross-domain variance in risk (e.g., Blais, A. R., & Weber, E. U. (2006). A Domain-Specific Risk-Taking (DOSPERT) scale for adult populations. Judgment and Decision

making, 1(1), 33-47.)” The authors do a good job of bringing in this important paper without confusing readers. To wit, the DOSPERT scale was developed to contrast differences between domains. Empirical work has subsequently shown that there is cross-domain shared variance, as Blaise and Weber acknowledge, but their emphasis and the whole point of their paper was that domains are different and that generalizing across domains is problematic. The authors navigate this potential confusion well.

In the future, although modeling metrics often combine goodness-of-fit with penalties for the number of parameters, the authors should separately consider how well each model fits the data and then whether added parameters are merited by the current data or external data. The overall metrics are mechanical, whereas a good theorist distinguishes less important and more important parameters and whether parameters (or assumptions) are needed to understand a phenomenon. No revisions are needed here to address this issue but it is something to keep in mind for future work. In sum, this is an impressive paper with relevance to fundamental issues of measurement and theory in decision-making.

**Have the authors made all data and (if applicable) computational code underlying the findings in their manuscript fully available?**

Reviewer #1: **No: ** Authors indicate data will be made public upon publication. Code is provided on the author's github (confirmed)

Reviewer #2: Yes

PLOS authors have the option to publish the peer review history of their article (what does this mean? ). If published, this will include your full peer review and any attached files.

**Do you want your identity to be public for this peer review?** For information about this choice, including consent withdrawal, please see our Privacy Policy .

Reviewer #1: No

Reviewer #2: No

**Figure resubmission:**
---

## [Editor Report · Decision Letter 2]

5 Feb 2025

Dear Dr. Korem,

We are pleased to inform you that your manuscript 'Modeling Decision-Making Under Uncertainty with Qualitative Outcomes' has been provisionally accepted for publication in PLOS Computational Biology.

Best regards,

Varun Dutt, Ph.D

Academic Editor

PLOS Computational Biology

Zhaolei Zhang

Section Editor

PLOS Computational Biology

---

## [Editor Report · Acceptance letter]

PCOMPBIOL-D-24-01428R2

Modeling Decision-Making Under Uncertainty with Qualitative Outcomes

Dear Dr Korem,

I am pleased to inform you that your manuscript has been formally accepted for publication in PLOS Computational Biology. Your manuscript is now with our production department and you will be notified of the publication date in due course.

With kind regards,

Zsofia Freund
